# Stabilizing A Vascularized Autologous Matrix with Flexible Magnesium Scaffolds to Reconstruct Dysfunctional Left Ventricular Myocardium in a Large-Animal Feasibility Study

**DOI:** 10.3390/jfb14020073

**Published:** 2023-01-29

**Authors:** Tobias Schilling, Serghei Cebotari, Tim Kaufeld, Igor Tudorache, Gudrun Brandes, Dagmar Hartung, Frank Wacker, Michael Bauer, Axel Haverich, Thomas Hassel

**Affiliations:** 1Department of Cardiothoracic, Transplantation, and Vascular Surgery, Medical School Hannover, 30625 Hannover, Germany; 2Institut National de Chirurgie Cardiaque et de Cardiologie Interventionelle, 1210 Luxembourg, Luxembourg; 3Department of Cardiac Surgery, Universitätsspital Zürich, 8091 Zürich, Switzerland; 4Institute of Neuroanatomy and Cell Biology in the Center of Anatomy, Medical School Hannover, 30625 Hannover, Germany; 5Institute for Radiology, Hannover Medical School, 30625 Hannover, Germany; 6IAV GmbH, 38518 Gifhorn, Germany; 7Institut für Werkstoffkunde (Materials Science), Leibniz Universität Hannover, 30823 Garbsen, Germany

**Keywords:** regenerative myocardial prostheses, transmural left ventricular reconstruction, cardiac surgery, magnesium degradation, magnesium scaffold

## Abstract

The surgical reconstruction of dysfunctional myocardium is necessary for patients with severe heart failure. Autologous biomaterials, such as vascularized patch materials, have a regenerative potential due to in vivo remodeling. However, additional temporary mechanical stabilization of the biomaterials is required to prevent aneurysms or rupture. Degradable magnesium scaffolds could prevent these life-threatening risks. A left ventricular transmural defect was reconstructed in minipigs with a piece of the autologous stomach. Geometrically adaptable and degradable scaffolds made of magnesium alloy LA63 were affixed on the epicardium to stabilize the stomach tissue. The degradation of the magnesium structures, their biocompatibility, physiological remodeling of the stomach, and the heart’s function were examined six months after the procedure via MRI (Magnetic Resonance Imaging), angiography, µ-CT, and light microscopy. All animals survived the surgery. Stable physiological integration of the stomach patch could be detected. No ruptures of the grafts occurred. The magnesium scaffolds showed good biocompatibility. Regenerative surgical approaches for treating severe heart failure are a promising therapeutic alternative to the currently available, far from optimal options. The temporary mechanical stabilization of viable, vascularized grafts facilitates their applicability in clinical scenarios.

## 1. Introduction

Several cardiovascular diseases, e.g., myocardial infarction, infectious diseases, or immunologic pathologies, cause the death of cardiomyocytes. Unlike many other tissues, the heart muscle has no regenerative potential. Deceased cardiomyocytes, thus, will be replaced by noncontractile connective tissue. If larger areas are affected, the formation of myocardial aneurysms and severe heart failure may develop. Surgical reconstruction of this dysfunctional myocardium is often the therapy of choice. In the 1980s, Dor et al. developed a surgical procedure to reconstruct damaged and dysfunctional myocardium, providing up-to-date basic surgical principles for this therapy [1]. After resection of nonfunctioning cicatricial tissue, the defect coverage is carried out either with synthetic materials, such as Dacron, or autologous pericardium. Since autologous pericardium is only available to a limited extent and Goretex is just as noncontractile as connective tissue, several biological tissues with a regenerative potential have been experimentally tested as a myocardial prosthesis [2]. The first results provide reason to hope that these biological tissues may be functionally integrated into the native myocardium due to an in vivo remodeling [3]. Most biological substrates are delicate materials and, therefore, require additional stabilization. Sufficient mechanical stability of a biological prosthesis is required, especially for the transmural reconstruction of the left ventricle, because a rupture of this tissue would immediately lead to the patient’s death. Such mechanical stability requires a minimum diameter of the patch tissue. Unfortunately, supplying oxygen and nutrients via diffusion becomes an issue in tissues with a thickness greater than 100 µm [4]. Thus, we employed a pre-vascularized stomach patch, including its natural vascularization, which ascertains the patch’s viability and, therewith, its regenerative potential. However, the stomach did not prove to be reliably stable enough. It sometimes forms an aneurysm under the high blood pressure of the left ventricle when used as a myocardial prosthesis in former experimental applications [5]. Degradable magnesium scaffolds could prevent this life-threatening risk to the patients. The scaffolds should be entirely degraded after the biological graft has achieved sufficient mechanical stability in vivo. Hence, magnesium scaffolds, in combination with regenerative biological grafts, could yield multiple advantages over conventional surgical reconstruction with nonviable patch materials. Though conventional and regenerative approaches both aim to restore the ellipsoid and hemodynamically efficient shape of the left ventricular myocardium [6], physiologically integrated grafts could actively contribute to the heart’s pump function and improve impaired cardiac performance. The magnesium scaffolds would temporarily provide the required mechanical stability for the graft until it gained strength through physiological remodeling and functional integration into the myocardium. 

A previous study has demonstrated, in both in vitro and simulations, that magnesium alloy LA63 structures, coated with magnesium fluoride and specifically adapted to the cardiac geometry, are less susceptible to premature fractures than flat shapes [7]. Hence, individual support structures should be pre-formed according to the respective cardiac geometry based on the patient’s CT or MRI (Magnetic Resonance Imaging) data to avoid plastic deformation. Developing segmented, flexible, and freely adaptable supporting structures made of degradable magnesium alloy would allow for individualized treatment for each patient. 

## 2. Materials and Methods

The intent was to cover a left ventricular full-wall defect with an autologous vascularized stomach patch in a chronic porcine model. The patch was to be temporarily stabilized with segmented, degradable scaffolds made of the magnesium alloy LA63. First, the clinical findings, such as survival rates, perioperative complications, and the movement and feeding behavior of the test animals, were be observed. Second, the heart’s pumping capacity was to be examined six months after surgery with MRI. Third, the explanted heart, including the hybrid graft, was to be investigated for biological and functional integration. Fourth, the degradation of the support structures was to be measured via µComputerTomography. The integrity and volume of the remaining structures was to be assessed. Fifth, the myocardium tissue’s resulting reaction and the stomach tissue’s physiological remodeling were to be examined histologically. The following research questions were posed: Are there any signs of graft rejection, foreign body reaction or other inflammation? Are there any signs of toxic effects of the metallic implants on the host’s cells, or of the biological integration of the cardiac and stomach tissues? 

### 2.1. Manufacturing Magnesium Scaffolds

The magnesium scaffolds were made from the magnesium alloy LA63, containing 6 wt% (weight percent) of lithium and 3 wt% aluminum. Extruded sheets of this alloy were cut with a thickness of 0.5 mm using abrasive waterjet technology to form a circular segmented structure. Garnet mesh #120 was used as the abrasive. The abrasive water jet cutting technique has already been described in detail elsewhere [8,9]. The structure’s design was previously calculated using finite element simulation [10]. The segmented support structures consisted of a central, oval ring (see Figure 1A), around which eight wing segments were placed in a circle (see Figure 1B). The central ring contained eight boreholes through which the wing segments were to be fixed pre-operatively immediately after sterilization with surgical sutures. At the central ring, the wing segments were secured with a nonabsorbable suture (Polyprolene 2.0, Ethicon, Germany). The wing segments were affixed to each other with an absorbable, braided suture (Vicryl 2.0, Ethicon, Germany) (see Figure 1). 

### 2.2. Implantation

All animal experiments were carried out in accordance with the European Convention on Animal Welfare and were approved by the licensing authority (LAVES, Lower Saxony) as per section 8, paragraph 1 of the Protection of Animals Act, German Civil Code 1. IS 01484 (experiment #08/1604). Animal surgery was performed according to the ARRIVE guidelines [11]. The anesthetic regimen and pre-operative and post-operative measures applied to prevent dysrhythmia, infections, pain, or gastric ulcers, as well as wound closure, have been described in detail elsewhere [9]. Lewe minipigs, weighing an average of 35 kg at the time of the operation, served as experimental animals in this series. We included only animals that showed no clinical signs of any pathologies. We carried out echocardiographic investigations of the test animals immediately before surgery. All animals exhibited physiological left ventricular ejection fraction values. All animals (n=6) received a left ventricular transmural reconstruction of an approximately 4 cm × 4 cm iatrogenic myocardium defect using a transdiaphragmatically transplanted stomach segment. The prepared piece of stomach contained the original vascular supply of the left epigastric artery and vein. The reconstruction was carried out with the autologous vascularized stomach segment without additional supporting structures for three of the animals meant to form the control group. In three other animals, the stomach was reinforced through geometrically adaptable and degradable magnesium alloy scaffolds consisting of nine separate (one central ring plus eight peripheral wings) and loosely connected segments, respectively (twenty-seven implanted segments in total).

An approximately 4 cm × 4 cm piece of stomach tissue with preserved vascularization was initially prepared via median laparotomy. The opened stomach was then closed with a continuous suture (Polyprolene 2.0, Ethicon, Germany). Subsequently, the laparotomy was extended to a left lateral thoracotomy in the fourth intercostal space to display the heart’s facies anterolateralis near the apex cordis. The heart surgery was performed after systemic heparinization (400 IU/kg; heparin sodium 25,000, Ratiopharm, Ulm, Germany) using a heart–lung machine (Stöckert S3, Sorin Group Germany GmbH, Munich) at a body temperature of 28 °C. The cannulation was placed in the carotid artery and the right atrium. In the Mg group, the prepared piece of the stomach, including the vascularization, was first guided through the central ring of the magnesium support scaffold (see Figure 2). Then, a 4 cm × 4 cm area of the left ventricular myocardium was transmurally resected in all animals. This defect was then closed with the stomach patch using single-button sutures (Polyprolene 4.0, Ethicon, Germany).

In the animals of the Mg group, nine segments of the magnesium scaffold were now epicardially fixed in the boundary zone between the myocardium and stomach tissue using single-button technology (Polyprolene 4.0, Ethicon, Germany) (see Figure 3). 

The antagonization of the heparin was carried out with 400 IU/kg of protamine (Medapharma, Switzerland) and heating at 37 °C body temperature to end the cardiopulmonary bypass. After careful hemostasis, the closure of the surgical site and the wound was performed via adaptation of the ribs (Mersilene 2.0, Ethicon, Germany), muscle layers (Vicryl 2.0, Ethicon, Germany), Donati cutaneous suture (CBX1 Vicryl, Ethicon, Germany), and wound sealing with aluminum spray (Almapharm, Germany).

### 2.3. Cardiac Magnetic Resonance Imaging (MRI)

All six test animals were examined by MRI six months after the surgical procedure, just before being euthanized. To immobilize the animals during the MR examination, they were sedated with intravenous Propofol Lipuro (1 mL/kg body weight/h, Braun, Melsungen, Germany) and then placed into the MRI scanner in the left lateral position. The MRI scan was performed in a 1.5 Tesla MR scanner (Avanto Magnetom, Siemens Healthcare, Erlangen, Germany). A four-element phased array receiver coil was used as a surface coil. The objective of the examination was to evaluate the morphology, function, and tissue characteristics of the left ventricle, particularly of the patch region. For the morphological and functional assessment of the heart, breath-stopped and ECG-gated cine true fast imaging steady-state precession (trueFISP) sequences were made in the heart standard levels. The following parameters were used to record the short axis stack through the left ventricle from the cardiac valve level to the apex: TR, 48.9 ms; TE, 1.37 ms; flip angle, 54 degrees; acquisition matrix, 256 × 208; slice thickness, 5 mm. In addition, two and four-chamber views were generated. To assess the possible fibrosis area, an inversion-recovery (IR) TrueFISP 2D sequence with the following parameters was used as delayed enhancement (DE): TR, 472 ms; TE, 3.3 ms; flip angle, 15 degrees; acquisition matrix, 256 × 150; slice thickness, 5 mm. The inversion time (IT) was adjusted so that the signal of the native myocardium was zeroed and was between 170 ms and 250 ms. The images were taken about 10–15 minutes after an intravenous gadolinium-containing contrast medium injection (Gadobutrol; Gadovist, Bayer Vital GmbH, Leverkusen, Germany) with a dose of 0.15 mmol/kg body weight. 

The quantitative analysis of the left ventricular volume and function data was performed with the software CVI42, version 5.1.2 (Circle Cardiovascular Imaging Inc., Calgary, Canada). For this purpose, the endocardial and epicardial contours in the end-systolic and end-diastolic phases of the short-axis stack were marked via the left ventricle from the atrioventricular junction to the cardiac apex. The left ventricle’s morphological characteristics and wall motion were assessed visually.

### 2.4. Explantation of the Heart, Including the Heterotopically Applied Segment of the Stomach and the Remains of the Supporting Magnesium Structure

The explantation of the heart and the grafts after six months of observation was performed under general anesthesia, which has already been described in detail elsewhere [9]. While the heart surgery for the transplantation of the gastric segment and implantation of the magnesium structures took place via left thoracotomy, the explantation was performed through a median sternotomy to preserve cardiac structures and the gastrointestinal transplant. Possible adhesions of the operated area with the visceral pleura could be spared using this access. After careful preparation of all adhesions, euthanasia of the test animals was performed with Phenobarbital (450 mg/kg body weight, WDT, Garbsen, Germany). The removal of the heart was carried out in the usual way, by severing through the aorta, vena cava superior and inferior, the pulmonary vessels, and, finally, the pedicle of the stomach patch.

The left ventricle was opened at the height of the interventricular septum, and the transplanted gastric segment was removed from the native myocardium with an approximately 1 cm margin.

### 2.5. Micro-Computed Tomography (μ-CT) Investigation

Before sterilization and implantation, the μ-CT investigation of the stabilizing magnesium alloy segments was executed. The investigation protocol has already been described elsewhere [9]. A total of 198 segments were investigated. A complete set, consisting of a central ring and 8 wings arranged radially around this ring, exhibited an average volume of 388.71 mm^3^ (standard deviation: 19.51 mm^3^).

After the six-month observation period, the explants that contained the remains of the degraded magnesium structure were analyzed by μ-CT. The resolution of the μ-CT was 36 microns per scanned layer. The data were reconstructed into a three-dimensional image with the software of the computer tomograph (Scanco Medical AG, Brüttisellen, Switzerland). Finally, the volumes of the remaining magnesium structures were calculated.

### 2.6. Histology

Immediately after completion of the μ-CT examination, all specimens were prepared for histologic evaluation. The embedding, cutting procedure, and staining have already been described in detail elsewhere [9]. The tissue was set in a formaldehyde solution (Otto Fischer GmbH & Co. KG, Weyarn, Germany) and embedded in paraffin (Carl Roth GmbH & Co. KG, Karlsruhe, Germany) for staining with Movat’s pentachrome (Waldeck GmbH & Co. KG, Münster, Germany) and then cut (Ultratome III 8800, LKB Bromma, Sweden).

## 3. Results

### 3.1. Clinical Results and Macroscopic Findings

All test animals survived the surgery. There were no significant complications in the post-operative phase. The animals showed normal movement and feeding behavior a few days after this highly invasive bi-cavity surgical procedure. In the open situs, an aneurysm of stomach tissue was seen in two cases in the control group and one in the Mg group. In these cases, a blood volume increase was recorded in the sac of the stomach tissue during the ventricular systole. The tissue became dyskinetic in these cases. Pronounced cicatricial changes were found in the border zone between the stomach transplant and the native myocardium (see Figure 4). In some places, the nonabsorbable sutures were seen in the fibrotic tissue (see Figure 4A). The implanted magnesium scaffold could not be detected on the epicardial surface (see Figure 4).

There was an incomplete closure of the transmural ventricular defect in the cases with aneurysmal changes of the transplanted gastric tissue (see Figure 5).

The edge of this gap consisted primarily of scarred connective tissue, demarcated from the native myocardium by its white color (see Figure 5). The typical pleated aspect of the muscle layer of the stomach tissue was visible through this opening (see Figure 5). Thrombotic deposits or signs of inflammation of the stomach tissue or the surrounding endocardium could not be seen in any case. There was a scarred closure of the ventricular defect in animals without an aneurysm of the stomach patch (see Figure 6).

### 3.2. MRI

All test animals were examined six months after successful surgery using cardiac MRI to evaluate the morphology and function of the left ventricle, particularly in the area of the transplanted patches of stomach tissue. In line with the macroscopic findings, stable integration of the stomach patch through adjacent cicatricial tissue could be detected in animals without an aneurysm (see Figure 7). 

The border zone between the myocardium and the stomach patch showed an intense late enhancement of the contrast agent as an expression of the adjacent stabilizing fibrotic remodeling processes in these animals (see Figure 8). 

However, the left ventricular ejection fraction was still reduced (50%), and the region of the stomach patch was akinetic. There was a paradoxical systolic outward movement of the affected area and a reduction of left ventricular ejection fraction of 15%–18% in the animals with an aneurysm. The aneurysms had a total volume of 12–18 mL. The delayed enhancement of the contrast agent was seen in these animals, mainly in the border zone between the native myocardium and the stomach tissue. 

### 3.3. μCT Investigation

The average volume of the magnesium scaffolds before implantation was 388.71 mm^3^ (standard deviation: 19.51 mm^3^). After 6 months, the remains of the degraded magnesium scaffolds in the explant, on average, possessed a volume of 180.73 mm^3^ (standard deviation: 8.79 mm^3^). Thus, the degradation in the observation period results in an average volume loss of 207.98 mm^3^ (standard deviation: 10.71 mm^3^). On average, a residual volume of magnesium of 46.49% remained in the explant 6 months after implantation.

More than half of the implanted volume of the magnesium alloy had disintegrated after six months. This volume loss was accompanied by considerable destruction of the segmented magnesium structure. Only granular fragments with an almost powdery consistency were visible in the μ-CT scans. Only a few wing segments were still recognizable as structures. Initially rather flat and edged, those segments now appeared round and distended (see Figure 9). The corrosion rate can be calculated to 0.052 g/month.

### 3.4. Histology

Histologically, excellent physiological integration of the heterotopic transplanted muscle layer of the stomach into the left ventricular myocardium appeared after six months following surgery. The myocardium appeared in the Movat’s pentachrome staining as reddish muscle cells with blue-black nuclei. The connective tissue in the border zone between the myocardium and stomach patch appeared yellowish and was penetrated by a cell-rich (blue-black-green) granulation tissue (see Figure 10). 

Within six months, all implants without an aneurysm had a similar thick layer consisting of collagen fibers and spindle-shaped cells. The granulation tissue and the stomach tissue were penetrated by numerous capillaries, which indicates a good integration of the vascularization of the stomach and myocardium. In all animals, the implanted gastric tissue was covered by a continuous layer of endothelial cells. Necrotic changes, signs of inflammation, calcific degeneration, or thrombus formation were not observed.

## 4. Discussion

The surgical treatment of severe heart failure with regenerative myocardial replacement materials would not only sustainably relieve the physical and psychological burden of the increasing number of affected patients [12], it would also mitigate the financial liability on healthcare systems from high medical expenses [13,14]. Our feasibility study demonstrated that transmural reconstruction with an autologous vascularized stomach is possible. Nevertheless, immediately after implantation, the stomach tissue is mechanically not stable enough to withstand the high blood pressure that prevails during systole in the left ventricle. Consequently, in our control group, an aneurysm of the heterotopic transplanted gastric segment was found in two of three animals. Stabilizing regenerative biological myocardial prostheses with degradable magnesium scaffolds should allow transmural left ventricular reconstruction with various promising biological matrices [15,16,17,18,19,20,21,22]. Those matrices have previously been used either in the low-pressure area of the right atrium and ventricle or only epicardially. In our study, all the animals, even those diagnosed with an aneurysm, survived the highly invasive bi-cavity procedure without substantial clinical complications. In the course of regular indoor breeding, there was no indication of impaired hemodynamics in any animal. 

The aspect of cicatricial tissue arose in the boundary zone between the gastric patch and the native myocardium. As seen in previous studies, the connection between the graft’s and heart’s vascularization and the synchronous contraction of the tissues indicates a functional integration [5]. Long-term side effects, such as ventricular arrhythmias due to an impaired electroconductive system, especially in the connective tissue of the boundary zone, could not be observed, which is in line with previous findings [5]. In animals with an aneurysm of the stomach tissue, there was a connection between the left ventricle and the stomach tissue, like a scarred channel. Hence, blood could be pumped into the lumen of the stomach patch through this connection during the ventricular systole. It remains questionable if a six months degradation time of the supporting magnesium alloy scaffolds is long enough. After six months, the scaffolds eventually could no longer be detected macroscopically. The μ-CT examination of the explants merely revealed granular to powdery clouds of the metallic implant, of which no significant mechanical stabilization function is expected. Perhaps the remodeling of the patch could benefit from a more extended degradation period. Moreover, covering a larger area of the healthy myocardium with the magnesium structures, in the sense of anchoring in more stable and healthy tissue, could improve the stabilizing function of the scaffold, which seems to be necessary for a physiological remodeling of the stomach tissue to an actively contractile patch without aneurysm formation.

The key advantage of magnesium as a substrate for temporary mechanical stabilization is its biocompatibility and well-known degradation process. Magnesium is an essential element in the body that plays important roles in biochemical reactions [23]. Hence, long-term side effects, such as ventricular arrhythmias, are not expected from the implantation of a magnesium-based scaffold. This work confirms magnesium scaffolds’ excellent biocompatibility, already shown in previous studies [9]. In addition, magnesium has no toxic effects in physiologic concentrations on other human body organs. However, we used the magnesium alloy LA63 to manufacture the stabilizing structures in this study. This alloy consists not only of magnesium but of 6 wt% lithium and 3 wt% aluminum. Feyerabend et al. found that lithium and aluminum have toxic effects on perivascular cells above a concentration of about 1000 µM [24]. The release of metallic ions in our experiments took place over a long period during the corrosion of the scaffolds. Hence, no unphysiologically high accumulation occurred in the organs primarily responsible for magnesium metabolism or its storage organs (skeletal muscle, liver, kidney, bone), as shown in previous studies [9].

The long-term control of the corrosion of the magnesium alloy in vivo thus appears to be the main challenge on the way to the clinical use of the support scaffolds for biological myocardial prostheses. The corrosion of magnesium in an aqueous solution results from the electrochemical oxidation of magnesium to Mg^2+^ and a reduction of water to hydrogen and hydroxide ions. Magnesium and hydroxide ions form hard soluble Mg(OH)_2_ [25,26]. Mg(OH)_2_ precipitates on the scaffold’s surface and hampers the corrosion. In solutions containing calcium, phosphates, and carbonates, such as body solutions, Mg(OH)_2_ converts into a hydroxide-like compound that is very efficient in protecting magnesium from rapid corrosion [27]. Due to the corrosion, irregularly large particles break away from the scaffold’s elements that disseminate into the surrounding tissue. They are further degraded as part of the foreign body reaction [28,29]. In addition to the corrosive attacks of the environment, the corrosion fatigue of metallic implants is dependent on the mechanical and shear stress’ frequency and strength acting on the implant [30,31]. Gu et al. determined a corrosion rate of up to 10 times higher for magnesium alloys when subjected to mechanical stress [32]. The cyclic action of the heart, which is not just a simple contraction and dilation, but rather a complicated process in which the myocardium wrings itself in a spiral, is just as much of a mechanical load as the high intraventricular blood pressure put on the biological graft. Mechanical stress on the magnesium scaffold was minimized by flexibly positioning the individual segments against each other by the loose connection of the surgical sutures. Thereby, the segments can adapt to the heart’s geometry and could thus dodge damaging pressure. Therefore, plastic deformation of the microstructure of the segmented scaffold used in this study is not presumed.

On the other hand, the cyclic action of the heart and the recurring displacement of the segments against the surrounding tissue may result in friction processes on the surface of the implants. Landolt et al. describe the exfoliation of surface components as wearing corrosion (tribocorrosion) [33]. The particles released through the exfoliation work the implant surface as abrasives. Witte et al. and Zheng et al. also hold shear stress responsible for the accelerated corrosion of magnesium implants when exposed to flow [34,35]. The enrichment of hydroxide ions and the associated alkalization of the environment would slow down the corrosion. However, in dynamic environments, such as in the epicardial localization presented in this study, the degradation products of the magnesium alloy, which primarily include magnesium hydroxide and hydrogen gas, but also include hydroxide ions, are quickly eliminated from the vicinity of the implant through biological and physical processes. Levesque et al. found a positive correlation between corrosion and flow rates [36]. 

Nevertheless, despite the corrosion of the magnesium alloy where, in addition to basically toxic products such as lithium or aluminum [24,37], hydrogen gas (H_2_) is also released, no adverse effects were seen on the biological integration and the anticipated remodeling processes. The volume of hydrogen gas of less than 1 ml/day that resulted over the long degradation period seemed to have diffused fast enough into its surroundings and was replaced by other gases, according to Kuhlmann et al. [38], so that no cytotoxic effects or an accumulation of a high volume of hydrogen gas were possible. The lack of toxicity of the implant and its degradation products in this study corresponds to the findings of a previous study in which we had determined the suitability of the magnesium alloy LA63 for application in the heart area [9]. The foreign body reaction to the metallic implant and the biological integration of the stomach transplant into the native myocardium architecture could proceed undisturbed through a cell-rich and well-capillarized granulation tissue. Due to its proven good biocompatibility, the magnesium alloy LA63 has the potential to demonstrate corrosion behavior that meets the requirements for stabilizing biological myocardial prostheses. The geometric adaptation of the support scaffolds to the patient’s or test animal’s individual anatomy continues to play an important role.

The present study may be interpreted as a feasibility study to assess the applicability of flexible magnesium scaffolds as stabilizers for biological grafts. Because of the small number of cases, statistical statements would not be valid. Future research should include larger series and focus on improving cardiac function, perhaps in myocardial infarction models. Such a model should thoroughly compare the heart’s pump function, e.g., according to the left ventricular ejection fraction, before the infarction, immediately after the infarction, and, eventually, after the transplantation of the transmural myocardial prosthesis. This study merely focused on the basic applicability of geometrically adaptable magnesium scaffolds to stabilize biological prostheses temporarily. Another limitation is the calculation of the volume loss of the magnesium structure using μ-CT examination. After the partial degradation of the magnesium structures in vivo, it is difficult to distinguish between the magnesium alloy and the degradation products around the magnesium hydroxide due to the similar radiopacity. Therefore, the calculation of volume loss by the degradation contains a particular uncertainty that must be considered. Finally, the reduced motility of the patch six months after implantation in some animals still limits the clinical application of this approach. 

## 5. Conclusions

The use of a vascularized stomach patch remains a promising substrate for a regenerative myocardial prosthesis because of the evidence of ingrowth of cardiomyocytes in gastric and intestine tissue [3], formation of a connection between the graft’s and heart’s vascularization, and, even, a link between the electrical system of the tissues resulting in synchronous contraction of the graft and the heart muscle [5]. The good clinical findings, the biological integration of the stomach, and the excellent biocompatibility of the magnesium scaffolds seen in this study indicate that autologous transplantation of a vascularized stomach segment is feasible to reconstruct a transmural left ventricular myocardium lesion. Stabilizing less durable but regenerative biological grafts with degradable magnesium scaffolds could allow for their application in a larger subpopulation of a growing group of patients with severe heart failure or ventricular aneurysms. Albeit, surgeons should continue using established materials, such as autologous pericardium and synthetic grafts, until regenerative left ventricular prostheses can be used clinically to reconstruct the left ventricular myocardium. A slower degradation rate of the magnesium scaffolds should be reached in future research to ascertain a more reliable stabilization and, consequently, an improved physiological remodeling with the ingrowth of cardiomyocytes into the biological component of the hybrid prosthesis. 

Even though the surgical approach introduced in this study is highly invasive, the severely impaired patients with terminal heart failure would benefit significantly from the regenerative, long-term, and curative therapy introduced in this study. It remains undisputed that regenerative surgical approaches for treating severe heart failure are needed to alleviate the mental and physical burden of patients for whom transplantation is not in sight due to the lack of donor organs. The increasing number of patients with congestive heart failure is a strong motivator to strive for regenerative therapeutic options, such as autologous vascularized matrices, that are in the pole position for future optimal myocardial prostheses. 

## Figures and Tables

**Figure 1 jfb-14-00073-f001:**
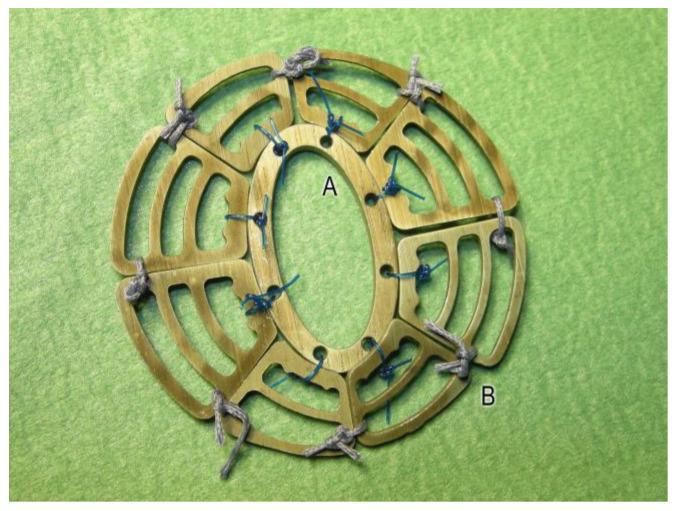
Nine immediately pre-operative connected segments of the flexible magnesium alloy LA63 scaffold. Central ring (A), circular wing segments of different sizes (B).

**Figure 2 jfb-14-00073-f002:**
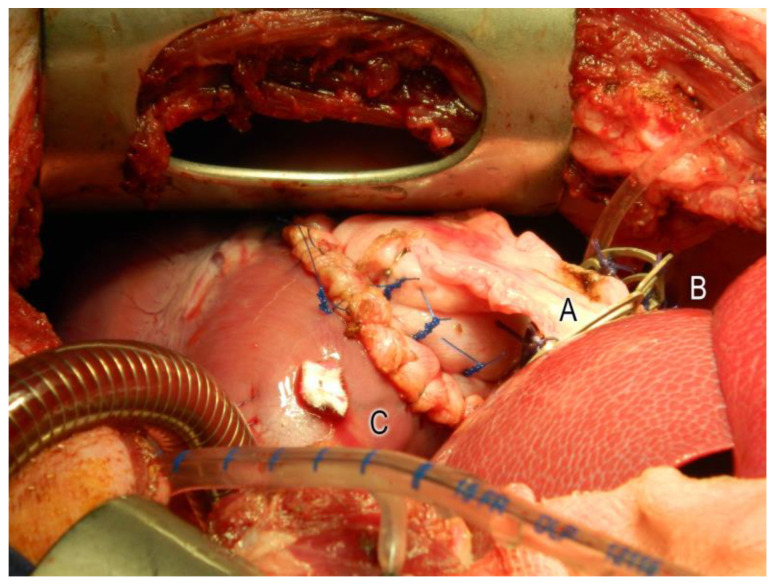
Operative situs: transdiaphragmatically transplanted stomach patch (A), fixed to the facies anterolateralis of the left ventricle (C) by single-button sutures. Guiding the pedicle of the stomach patch through the central ring of segmented magnesium alloy scaffold in the Mg group (B).

**Figure 3 jfb-14-00073-f003:**
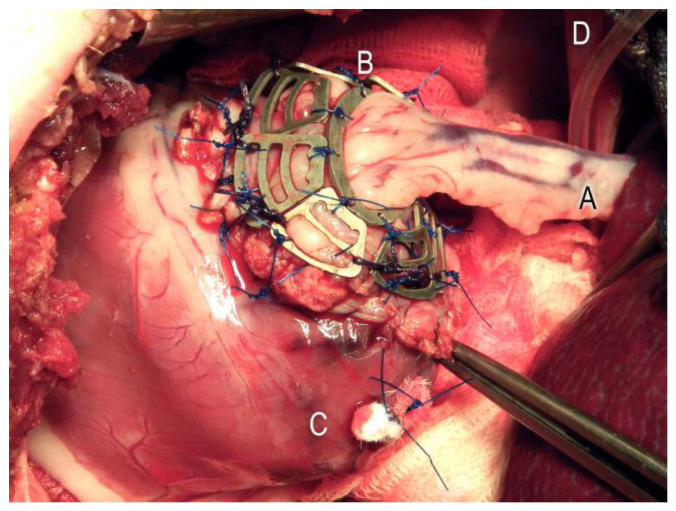
Nine epicardially fixed Mg segments (B) to stabilize the stomach patch (A) used to close the left ventricular (C) myocardium defect. Diaphragm (D).

**Figure 4 jfb-14-00073-f004:**
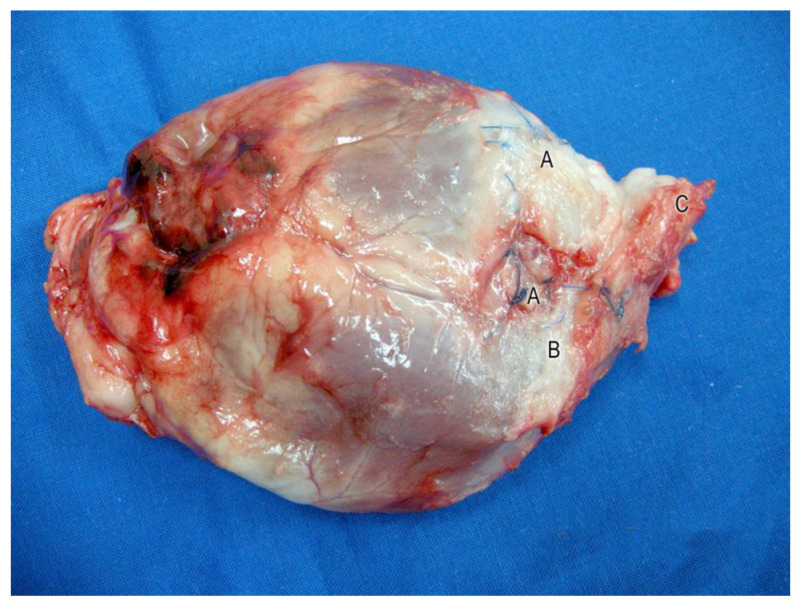
Explanted heart, six months following transplantation of autologous stomach and implantation of stabilizing magnesium scaffolds. Nonresorbable sutures (A) are seen under white, fibrotic connective tissue (B). The transplanted stomach patch pedicle (C) was cut just before passing through the diaphragm.

**Figure 5 jfb-14-00073-f005:**
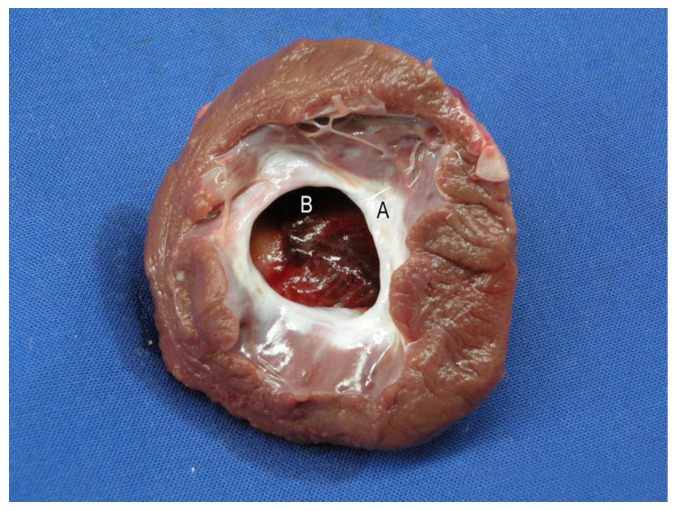
Luminal view of the opened left ventricle. Native myocardium is demarcated from the stomach patch by a whitish cicatricial border zone (A) in animals with aneurysm of the gastric tissue. The incomplete closure of the transmural ventricular defect facilitates the view of the typical pleated aspect of the stomach’s mucosa (B).

**Figure 6 jfb-14-00073-f006:**
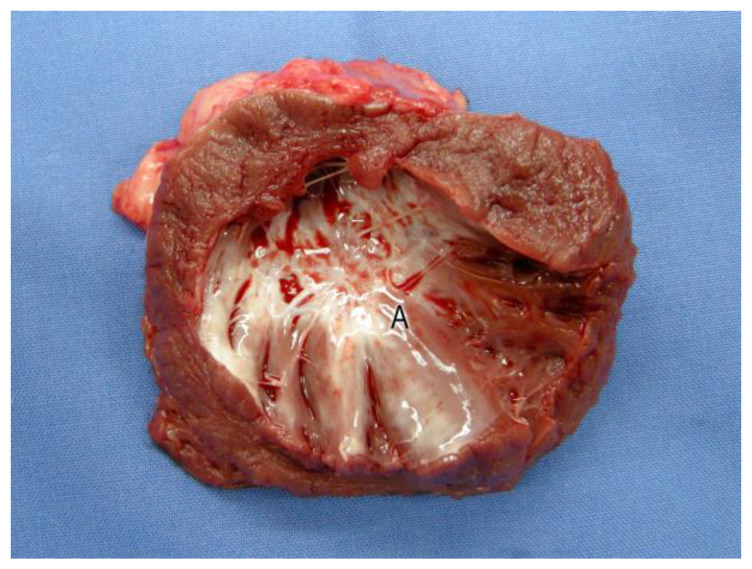
Luminal view on the opened left ventricle. Native myocardium is demarcated from the stomach patch (A) by a whitish cicatricial border zone. The complete closure of the transmural ventricular defect is noted in animals without an aneurysm of the stomach patch.

**Figure 7 jfb-14-00073-f007:**
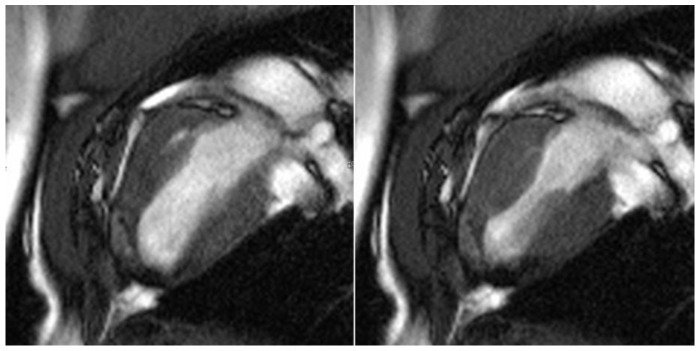
Cardio MRI six months following myocardial reconstruction with a vascularized stomach patch, stabilized with degradable magnesium alloy scaffolds. Cine SSFP in left ventricular two-chamber view. **Left**: end-diastolic phase, **right**: end-systolic phase. Good integration of the transplanted stomach patch.

**Figure 8 jfb-14-00073-f008:**
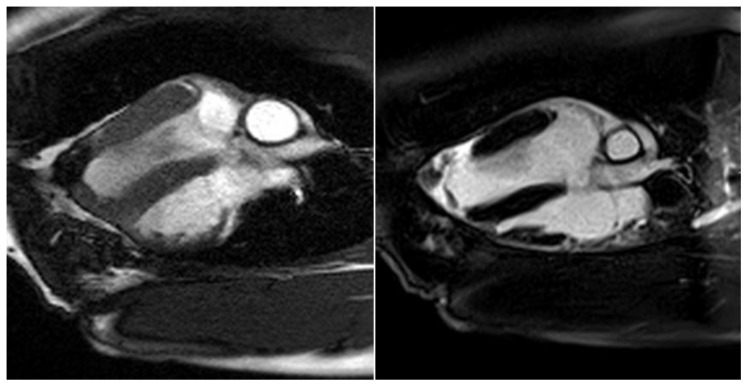
Cardio MRI six months following myocardial reconstruction with a vascularized stomach patch, stabilized with degradable magnesium alloy scaffolds. Delayed enhancement imaging. **Left**: cine SSFP. **Right**: four-chamber view with signs of distinct late enhancement in the left ventricular, apical zone, indicating cicatricial remodeling.

**Figure 9 jfb-14-00073-f009:**
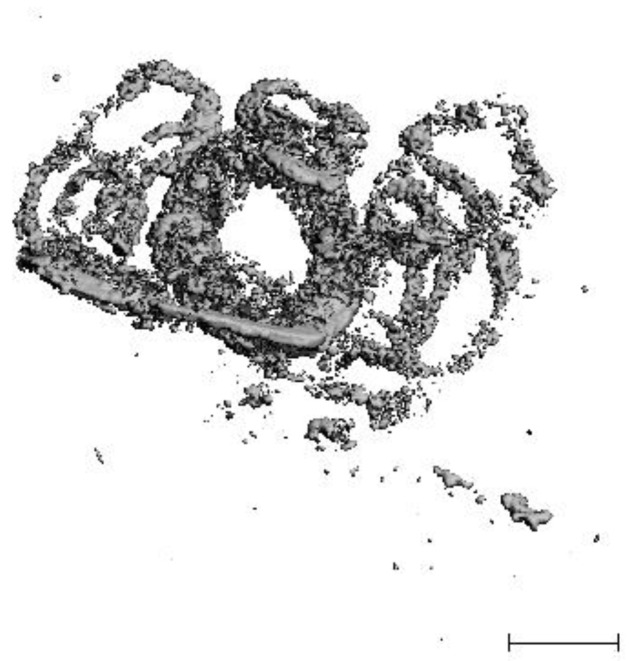
Reconstruction of µ-CT data of explanted tissue with residual magnesium scaffolds six months following implantation. Bar indicates 5 mm.

**Figure 10 jfb-14-00073-f010:**
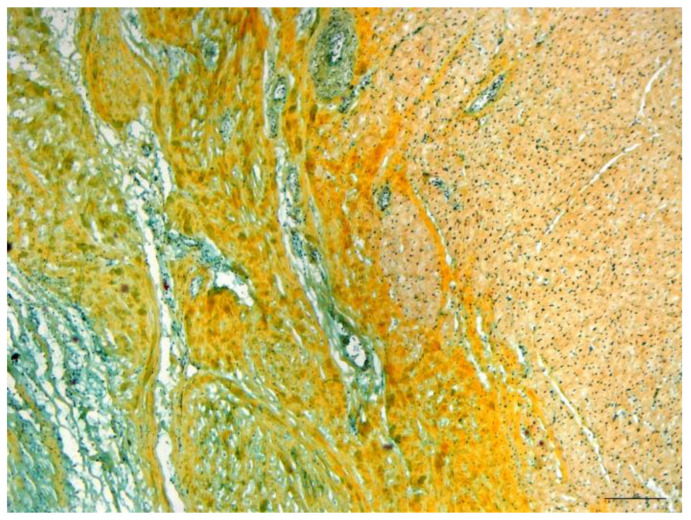
Gastric patch with stabilizing magnesium scaffold. Border zone between gastric patch and myocardium. Pentachrome staining. Bar indicates 200 µm.

## Data Availability

Not applicable.

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
