# Peer review of "Stabilizing A Vascularized Autologous Matrix with Flexible Magnesium Scaffolds to Reconstruct Dysfunctional Left Ventricular Myocardium in a Large-Animal Feasibility Study"

_jfb, 2023, doi:10.3390/jfb14020073_

Round 1

Reviewer 1 Report

Comments to the authors

In this interesting paper report an initial experience of surgical reconstruction of dysfunctional myocardium using a biological substrate on animals. In details, they used a transdiaphragmatically transplanted stomach segment containing the original vascular supply of the left epigastric artery and vein on six mini pigs. In three animals, the stomach was reinforced through geometrically adaptable and degradable magnesium alloy scaffolds (Mg-group). All animals survived the surgery. Nevertheless, an aneurysm of stomach tissue was seen in two cases in the control group and one in the Mg-group. The magnesium scaffold could not be detected on the epicardial surface.

I think the authors made a huge effort addressing an open issue such as the surgical reconstruction of myocardium with biological materials in the era of donor shortage.

Anyway I do have some concerns:

·      The authors used an autologous vascularized stomach patch, stabilized or not with the magnesium scaffold. Please explain why you chose the stomach tissue.

·      You state that all animals survived surgery and that there were no significant complications in the post-operative fase. However, this kind of surgery in pretty invasive because you first performed a median laparotomy and then a left thoracotomy. Do you think this degree of invasiveness would be justified if you look at the results?    

·      Line 253-254. “However, the left ventricular ejection fraction was still reduced (50%), and the region of the stomach patch was akinetic.” Unfortunately, I don’t think we can expect anything better from the use of a non-contractile tissue, even if it is a biological one. Please explain why we should prefer this to the standard Goretex patch. 

·      At the moment of the heart explantation, authors found an aneurysm of stomach tissue in two cases in the control group and one in the Mg-group, and an incomplete closure of the transmural ventricular defect in the cases with aneurysmal changes of the gastric tissue. They didn’t identify the cause of the persistence of this gap, but they suggest that it could be due to the poor stability of the stomach tissue immediately after surgery that could be improved by the use of the magnesium scaffold (Line 380-385). Anyway, I think that the authors do not have enough data to even suggest this answer. However, I think they should be encouraged to continue their work because regenerative surgical approaches might represent the future for heart failure treatment. 

Author Response

Dear Reviewer,

Thank you very much for your positive feedback and valuable comments on our manuscript. We complied with all your modification requests, as explained in detail below.

In this interesting paper report an initial experience of surgical reconstruction of dysfunctional myocardium using a biological substrate on animals. In details, they used a transdiaphragmatically transplanted stomach segment containing the original vascular supply of the left epigastric artery and vein on six mini pigs. In three animals, the stomach was reinforced through geometrically adaptable and degradable magnesium alloy scaffolds (Mg-group). All animals survived the surgery. Nevertheless, an aneurysm of stomach tissue was seen in two cases in the control group and one in the Mg-group. The magnesium scaffold could not be detected on the epicardial surface.

I think the authors made a huge effort addressing an open issue such as the surgical reconstruction of myocardium with biological materials in the era of donor shortage.

=> Thank you for your positive estimation of the work's impact.

Anyway I do have some concerns:

The authors used an autologous vascularized stomach patch, stabilized or not with the magnesium scaffold. Please explain why you chose the stomach tissue.

Patch materials with a thickness greater than 100 µm can not be supplied via diffusion and, thus, require vascularization. So, the main reason we chose the stomach as patch material is the possibility of transplanting an autologous vascularized segment of this organ, including its native vascularization. This pre-vascularization ascertains the patch's viability and, therewith, its regenerative potential. We added this information in the introduction section.

You state that all animals survived surgery and that there were no significant complications in the post-operative fase. However, this kind of surgery in pretty invasive because you first performed a median laparotomy and then a left thoracotomy. Do you think this degree of invasiveness would be justified if you look at the results?

A bi-cavity procedure is indeed a highly invasive surgery. We also hoped for a full physiological remodeling in-vivo in the observation period, as seen in clinical cases of right atrial reconstruction, which could be achieved in part in this study. But, the burden of congestive heart failure, including severe physical and psychological issues of many patients worldwide calls for new, long-term, and not least, curative therapeutic measurements. The current surgical options, such as the Dor procedure, cardiac transplantation, and especially the implantation of left ventricular assist devices (LVAD) are not much less invasive and provide dissatisfactory results. Even though we can report individual patients who survive more than 10 years on LVAD, most LVAD patients suffer from this therapy's side effects, such as life-long anticoagulation therapy, risk of infection, impaired mobility, and constant psychological stress. Despite modern immuno-suppression treatment, cardiac transplants still are susceptible to graft rejection of the host. And the risk of losing the graft is especially threatening in the face of the lack of donor organs. Conclusively, yes, the highly invasive approach introduced in this study is definitely justified because our procedure aims to mitigate the risks of life-threatening adverse events of the available options, increase the quality of life of long-term patients, and decrease the immense costs of current standard therapies.

Thank you for this impulse. Your question inspired us to emphasize this aspect in the “Conclusions” section.

Line 253-254. "However, the left ventricular ejection fraction was still reduced (50%), and the region of the stomach patch was akinetic." Unfortunately, I don't think we can expect anything better from the use of a non-contractile tissue, even if it is a biological one. Please explain why we should prefer this to the standard Goretex patch.

If the remodeling of the biological graft would not take place and the stomach would not contribute to the heart's pump function, biological and regenerative prostheses would indeed not have a functional advantage over synthetic materials. In this case, their benefit could be reduced to, at least, less risk or better manageability of endocarditis, e.g. However, since we found ingrowth of cardiomyocytes in similar grafts if used for reconstruction of the atrial wall, a good connection between the graft`s and heart`s vascularization and even a link between the electrical system of the tissues resulting in synchronous contraction of the graft and the heart muscle in previous studies, the use of a vascularized stomach patch still is a promising substrate for a regenerative myocardial prosthesis. Thus, a prolonged degradation period of the magnesium structures and probably covering a larger area of the healthy myocardium in the sense of anchoring in more stable and healthy tissue is perhaps required for a reliable and clinical left ventricular application because stabilizing the biological graft seems necessary for a physiological remodeling to an actively contractile patch without aneurysm formation.

We are grateful for this impulse, which sheds light on the experimental character of our study. Surgeons should indeed continue using established materials, such as autologous pericardium and synthetic grafts before the clinical application of regenerative left ventricular prostheses can be carried out. Nevertheless, we should strive for regenerative therapeutic options like autologous vascularized matrices. We added this perspective to the “Conclusion” part, too.

At the moment of the heart explantation, authors found an aneurysm of stomach tissue in two cases in the control group and one in the Mg-group, and an incomplete closure of the transmural ventricular defect in the cases with aneurysmal changes of the gastric tissue. They didn't identify the cause of the persistence of this gap, but they suggest that it could be due to the poor stability of the stomach tissue immediately after surgery that could be improved by the use of the magnesium scaffold (Line 380-385). Anyway, I think that the authors do not have enough data to even suggest this answer.

The aneurysm was certainly a consequence of insufficient mechanical stability of the gastric patch. Aneurysms form because of a mismatch between the internal blood pressure and the vessel wall's mechanical resilience. So, this conclusion seems plausible. Nevertheless, it is true that the hypothesis of the remaining gap in the ventricle wall due to an oscillating blood volume is merely a hypothesis without any underlying data. So, we restricted this part of the discussion to describing the phenomenon and the possible correlation between aneurysm formation and the graft's insufficient mechanical stability.

However, I think they should be encouraged to continue their work because regenerative surgical approaches might represent the future for heart failure treatment.

Thank you for your encouragement. We are also convinced that the future of treating multiple cardiovascular diseases lies in regenerative implants and fiercely work on engineering better solutions for our patients.

Happy new year and all the best

Tobias Schilling, on behalf of the manuscript's authors

Reviewer 2 Report

1) It would be useful to discuss the potential applicability of flexible magnesium scaffolds in left ventricular aneurysm surgical exclusion (e.g., following myocardial infarction);

2) Advantages and disadvantages of flexible magnesium scaffolds versus standard surgical myocardial reconstruction techniques should be appropriately discussed. Also, some clinical and imagistic features favoring magnesium scaffolds implantation should be provided;

3) Investigated outcomes should be defined and described in the methods section;

4) The authors stated: “However, the left ventricular ejection fraction was still reduced (50%), and the region of the stomach patch was akinetic” – left ventricular ejection fraction prior and after the intervention should be provided;

5) Potential long-terms complications of magnesium scaffolds should be discussed (including ventricular arrhythmias);

6) Some directions and drawbacks of future research (including on humans) should be identified and discussed;

7) Following the discussion section, the authors should provide some conclusions, reflecting the main outcomes of the study.

Author Response

Dear Reviewer,

Thank you very much for your positive feedback and your valuable comments on our manuscript. We complied with all your modification requests as explained in detail below.

1) It would be useful to discuss the potential applicability of flexible magnesium scaffolds in left ventricular aneurysm surgical exclusion (e.g., following myocardial infarction);

We tried to keep the introduction quite brief; thus, we summarized the pathologies under the term "heart failure." Your comment indicated though that a more detailed description of heart failure pathology could add value to this chapter. The surgical therapy of aneurysms requires patch materials that ideally should have a regenerative capacity. Hence, stabilizing those grafts with magnesium scaffolds would support the reconstructive operation. According to this surgical approach, magnesium scaffolds are always only a temporary supporting measure until a physiological remodeling of the biological graft has taken place. Thank you for this advice. We extended the chapter and the wording accordingly.

2) Advantages and disadvantages of flexible magnesium scaffolds versus standard surgical myocardial reconstruction techniques should be appropriately discussed. Also, some clinical and imagistic features favoring magnesium scaffolds implantation should be provided;

Since magnesium scaffolds are merely a supporting system for the biological grafts, their advantages and disadvantages should always be regarded for the entire hybrid graft consisting of biological patch material and stabilizing magnesium scaffolds. We emphasized this aspect a bit stronger in the introduction section.

According to your request, we extended the discussion and emphasized the magnesium scaffolds' good applicability in-vivo, which is the key feature and rationale for their use in this study. We also indicated some possible disadvantages of magnesium alloys as cardiovascular implants due to the potential cytotoxic effects of the alloying elements lithium and aluminum.

3) Investigated outcomes should be defined and described in the methods section;

Done. We already introduced the basic experimental design in the last paragraph of the introduction chapter but have moved this to the methods section now. Moreover, we described the investigated outcomes in this paragraph.

4) The authors stated: "However, the left ventricular ejection fraction was still reduced (50%), and the region of the stomach patch was akinetic" – left ventricular ejection fraction prior and after the intervention should be provided;

For future applications, especially in clinical settings, the targeted long-term improvement of the heart's function, represented amongst others by the left ventricular ejection fraction, is undoubtedly of paramount relevance for the surgical concept's success. Future studies, therefore, should not only span larger animal numbers to build a solid statistical fundament for the first in men cases. They should also thoroughly assess the pump function before the intervention and compare those data with short-, mid-, and above all, long-term results. The study at hand was designed as an introductory feasibility study to assess if flexible magnesium scaffolds are basically applicable in left ventricular reconstruction. We included only animals that showed no clinical signs of any pathologies. Nevertheless, we carried out echocardiographic investigations of the test animals immediately before surgery and only found physiological ejection fraction values. However, we felt that we should not make an explicit comparison of echocardiographic analyses with the post-operative MRI-based values due to the differences in the methodologies' precision. The missing comparison can be seen as a further limitation of this study, which should be addressed in upcoming tests. We added this aspect to the "Limitations" section and set this as a direction for future research, also according to your 6th impulse (s. below). Moreover, in the methods section, we mentioned the basic echocardiographic, pre-operative investigation and its typical results as a baseline.

5) Potential long-terms complications of magnesium scaffolds should be discussed (including ventricular arrhythmias);

We employed fully degradable magnesium alloy scaffolds aiming for a complete absence of any magnesium residues after the scaffolds` stabilization function is no longer needed. Hence, we do not expect any long-term complications from the magnesium or its degradation process. We assessed the biocompatibility of the magnesium alloy in a previous study. There, we did not find any toxic effects on the surrounding cardiac tissue or the host's magnesium storage organs, such as the bone, liver, spleen, etc.. Magnesium is an essential element with no known side effects in physiological conditions. Nevertheless, our scaffolds and their degradation products activate the normal foreign body reaction. Also, magnesium degradation leads to the release of hydrogen gas, which can be problematic in, e.g., bone surgery, the thoracic cavern, especially due to the opened pericardium sac, offers enough space to evacuate the gas. Thus, we could not observe the accumulation of hydrogen gas at all. Second, we did not find evidence of a toxic accumulation of lithium and aluminum alloying elements in our previous biocompatibility study, nor did we observe any regional or systemic harmful effects of these elements. We discussed and mentioned these aspects in the discussion but extended the perspective on the (not-)expected long-term effects according to your request in the discussion part.

6) Some directions and drawbacks of future research (including on humans) should be identified and discussed;

We stated the clinicals implications of the current limitations of the study's findings in the newly created "conclusions" chapter (s. 7.).

7) Following the discussion section, the authors should provide some conclusions, reflecting the main outcomes of the study.

We added a sub-chapter "conclusions" in the discussion section and moved and elaborated our conclusions there.

Happy new year and all the best

Tobias Schilling, on behalf of the manuscript's authors

Reviewer 3 Report

I would like to congratulate the authors for this novel feasibility study on presenting potential regenerative surgical approaches for treating severe heart failure. I have no major changes to recommend and would like to accept the paper in current form.

Author Response

Dear Reviewer,

Thank you very much for your positive feedback and your valuable comments on our manuscript. We are happy that you have accepted our manuscript.

Happy New Year and all the best.

Tobias Schilling

Round 2

Reviewer 1 Report

Dear authors,

thank you for the accurate answers to all my questions.

I don't have any other concern about the paper and I think it is suitable for publication.

Good luck on your work!

Reviewer 2 Report

It seems ok after revision

The manuscript could be published in this form.